# Immunohistochemistry in an Adult Case of Bitot’s Spots Caused by Vitamin A Deficiency

**DOI:** 10.3390/diagnostics13243676

**Published:** 2023-12-15

**Authors:** Hideki Fukuoka, Norihiko Yokoi, Chie Sotozono

**Affiliations:** Department of Ophthalmology, Kyoto Prefectural University of Medicine, Kyoto 602-8566, Japan; nyokoi@koto.kpu-m.ac.jp (N.Y.); csotozon@koto.kpu-m.ac.jp (C.S.)

**Keywords:** vitamin A deficiency (VAD), Bitot’s spots (BS), keratinization, and amniotic membrane transplantation (AMT)

## Abstract

Bitot’s spots (BS) are the buildup of superficially located keratin in the conjunctiva and are early indicators of vitamin A deficiency (VAD), primarily due to malnutrition and malabsorption, thus leading to xerophthalmia. BS are particularly prevalent in developing countries, and their presence necessitates prompt vitamin A supplementation to avert blindness, with the immunohistochemical characteristics of BS aiding in understanding the extent of epithelial abnormalities and the efficacy of vitamin A supplementation. We describe the case of a 34-year-old male with persistent BS despite extensive vitamin A supplementation and topical treatments who underwent surgical excision of the BS followed by amniotic membrane transplantation, thus resulting in symptom relief and epithelialization, with no recurrence observed during follow-up. Histopathologic and immunohistochemical evaluations revealed expression of keratinization-related proteins, along with an absence of mucin-5AC-positive cells, suggesting impaired differentiation into goblet cells due to VAD. This case highlights the potential age-related disparity in the efficacy of vitamin A supplementation, emphasizing the need for early detection and a multidisciplinary approach in the management of VAD, especially in young adults. The favorable outcome of surgical intervention highlights its viability in the management of persistent BS and encourages further investigation to optimize therapeutic strategies for VAD-related ocular manifestations.

**Figure 1 diagnostics-13-03676-f001:**
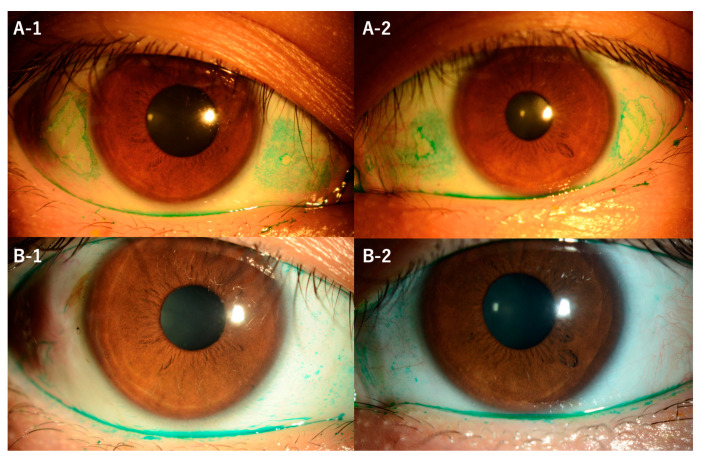
A 34-year-old east Asian (Japanese) male with bilateral ocular dryness and foreign body sensation was ultimately referred to our hospital for further evaluation after a 1-year treatment with a P2Y2 receptor agonist (diquafosol), rebamipide, and betamethasone eye drops for allergic conjunctivitis and dry eye at an outside clinic was unsuccessful for relieving the symptoms. Upon initial examination, his bilateral best-corrected visual acuity was better than 20/20; however, Bitot’s spots (BS) were observed on the nasal and temporal conjunctiva in both eyes (**A-1**,**A-2**). Laboratory tests revealed a serum vitamin A level of 80 IU/dL (normal range: 97–316 IU/dL), thus confirming vitamin A deficiency, and investigation into the patient's dietary history revealed a significant lack of vegetable consumption due to the coronavirus disease 2019 (COVID-19) pandemic lockdown. For treatment, the patient was administered 30,000 IU of vitamin A along with rebamipide eye drops four times daily, levofloxacin eye drops, and low-dose steroid eye drops twice daily. At the 4-month follow-up visit, there was no improvement in BS despite the serum vitamin A level having increased to 120 IU/dL. Subsequently, surgical excision of the BS followed by amniotic membrane transplantation was performed. Post-surgery, epithelialization occurred, the patient's symptoms subsided, and no recurrence of the BS was observed throughout the subsequent follow-up period. Preoperative impression cytology of the conjunctiva including BS and excision specimens were subjected to histopathologic and immunohistochemical evaluation. In the preoperative impression cytology of the area of the BS, it was not possible to obtain tissue directly from the BS, and only tissue from other areas could be obtained. Slit-lamp microscopy images of the patient's right (left-side column) and left (right-side column) eyes showed elevated BS stained with Lissamine Green before (top row) and after (bottom row) surgical excision of the BS and amniotic membrane transplantation. Preoperative images showing Lissamine Green staining revealed elevated BS located nasally and temporally in both eyes, with the temporal lesions being larger than the nasal ones (**A-1**,**A-2**). Postoperative images showed complete resolution of the BS (**B-1,B-2**).

**Figure 2 diagnostics-13-03676-f002:**
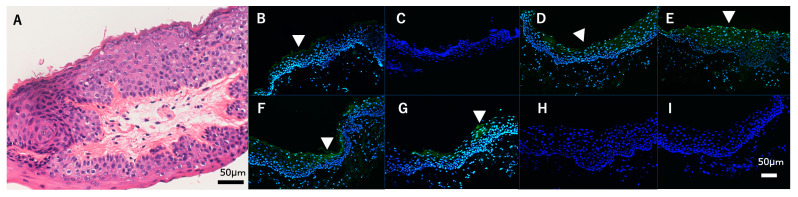
Hematoxylin and eosin-stained pathological tissue specimen of an excised Bitot’s spots (BS) showing squamous metaplasia and keratinization in the superficial layers. The type of keratinization observed is hyperkeratosis, as evidenced by the retention of nuclei in the keratinized cells. (**A**) Scale Bar = 50 μm. Pathology showed squamous epithelialization and parakeratosis of the superficial epithelium (Figure 2). Images showing immunofluorescent results against cytokeratins, keratinization, and mucin (MUC) markers of the obtained tissue (**B**–**I**). (**B**) Cytokeratin (CK) 1, (**C**) CK10, (**D**) CK4, (**E**) CK13, (**F**) transglutaminase-1, (**G**) filaggrin, (**H**) involucrin, and (**I**) MUC5AC (green) and DAPI for counter-staining (blue) for the BS. Bar = 50 μm. BS tissues were collected during surgery and embedded in Tissue-Tek^®^ O.C.T. Compound (Sakura Finetek USA, Inc., Torrance, CA, USA), then snap-frozen in liquid nitrogen. Next, 8 μm thick frozen sections were prepared on slides, air-dried, and fixed in Zamboni’s fixative, and then washed in 0.01M phosphate-buffered saline. The sections were then blocked with 1% bovine serum albumin before incubating overnight at 4°C with primary antibodies, followed by Alexa Fluor 488 conjugated secondary antibodies and once again being washed. Finally, the sections were imaged using a fluorescence microscope. In the conjunctival impression cytology, it was not possible to directly retrieve epithelial cells covering the BS, which were negative for mucin-5AC (MUC5AC). Immunohistochemical analysis showed that BS pathological tissue expressed CK1, CK4, CK13, and keratinization-related proteins (i.e., transglutaminase 1 and filaggrin) (white arrow heads) and was negative for CK10, loricrin, and MUC5AC (Figure 2). BS are characteristic lesions of the buildup of superficially located keratin in the conjunctiva and are early indicators of vitamin A deficiency (VAD) [1], a condition that can lead to blindness as vitamin A is critical for normal differentiation and proliferation of human conjunctival epithelium. The primary underlying cause of VAD is malnutrition, followed by malabsorption, which can lead to a variety of ocular manifestations collectively known as xerophthalmia. BS are specific early signs of VAD and present as dry, whitish, foamy, oval, triangular, or irregular-shaped lesions on the conjunctiva, primarily on the temporal side of the eye, and they consist mainly of keratin mixed with the gas-forming bacteria Corynebacterium xerosis, which gives them their foamy appearance [2]. In eyes afflicted with BS, early intervention with vitamin A supplements can reverse the condition and stop the progression to blindness. However, if left untreated and xerophthalmia and blindness develop, mortality rates can be as high as 40% within the first year of diagnosis [3]. In addition, the prevalence of BS in children is often used to assess the burden of vitamin A deficiency, particularly in developing countries where malnutrition is prevalent [4]. Vitamin A is critical for cell differentiation and affects genome expression by regulating certain gene transcriptions, modifying mRNA levels, and altering membrane structure and function. Moreover, it affects protein biosynthesis, including proteins essential for development and cell function, and influences the formation of hormone-like secretory proteins. VAD disrupts these processes and leads to differentiation disorders and conditions such as BS [2]. The case presented in this report highlights the persistence and severity of VAD, particularly in young adults, and the resulting therapeutic challenges. In contrast to younger individuals in whom vitamin A supplementation [3,5] often results in a rapid resolution of the symptoms, our patient, a 34-year-old male, had a more refractory course. Despite extensive vitamin A supplementation along with topical treatments, the BS—a hallmark of VAD—remained refractory, highlighting a potential age-related disparity in therapeutic outcomes. We believe that prolonged vitamin A deficiency remains the most likely primary cause underlying the persistent BS in this case. Even with supplementation, the turnover and replacement of his markedly affected conjunctival tissue are both difficult and slow. Therefore, we suspect that refractory BS have persisted due to the severity and duration of his long-standing deficiency rather than genetics or other factors. Although surgical excision of the BS followed by amniotic membrane transplantation proved to be a highly effective solution, there are currently no published reports on the efficacy of amniotic membrane transplantation for BS. In the present case, postoperative epithelialization occurred, leading to a significant improvement in the symptoms, thus highlighting the potential role of surgery as a viable option in the management of persistent BS in young adults. In addition, the histopathologic and immunohistochemical evaluations provided valuable insights into the pathology of BS. The presence of CK4 and CK13 derived from mucosal tissues emphasizes that the epithelium retains mucosal properties [6]. Interestingly, the detection of CK1, a protein typically expressed in skin tissues [7], and the keratinization-related proteins transglutaminase-1 and filaggrin highlight a remarkable degree of keratinization. The absence of involucrin [8] further delineates the state of partial keratinization, characteristic of defective keratinization such as parakeratosis, possibly driven by VAD [9]. The absence of MUC5AC-positive cells in preoperative conjunctival impression cytology, including BS and excised specimens, suggests that VAD may impair differentiation into goblet cells [10,11,12]. In the published literature, there is evidence that retinoid signaling regulates goblet cell differentiation in the eye, including the findings in the recent report by Alam et al. that showed decreased conjunctival goblet cells in vitamin A deficiency and attenuated goblet cell numbers with disrupted RXRα signaling [13]. Those findings demonstrate that retinoid signals play an indispensable role in ocular surface goblet cell homeostasis. The case presented in this report highlights the critical role of early detection and multidisciplinary management in mitigating the sequelae of VAD. The differential response to vitamin A supplementation between younger individuals and young adults requires further investigation to elucidate the underlying mechanisms and optimize therapeutic strategies. It should be noted that the findings in this report are consistent with the existing literature describing the interplay between vitamin A, epithelial integrity, and ocular surface disorders, including dry eye and superior limbic keratoconjunctivitis. The nuanced understanding of the pathological changes and the efficacy of surgical intervention in this present case contribute to the broader discourse on the management of ocular manifestations of VAD. The present case highlights the complexity of treating VAD in young adults and demonstrates a potential age-related disparity in outcomes. Surgical intervention with excision of BS and amniotic membrane transplantation proved effective in alleviating symptoms, highlighting the need for a multidisciplinary approach and further investigation to optimize therapeutic strategies. In regard to the immunohistochemistry of BS, further research is required, as the scientific information has yet to be elucidated in detail, i.e., the cellular and molecular aspects of BS and their interaction with vitamin A at the tissue level would provide a deeper understanding of the pathology and mechanisms underlying this condition. In addition, further investigation into the immunohistochemical characteristics of BS may shed light on the extent of epithelial abnormalities and the efficacy of vitamin A supplementation in reversing those abnormalities.

## Data Availability

The data presented in this study are available upon request from the corresponding author.

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
