# Peer review of "Immunohistochemistry in an Adult Case of Bitot’s Spots Caused by Vitamin A Deficiency"

_diagnostics, 2023, doi:10.3390/diagnostics13243676_

Round 1
Reviewer 1 Report
Comments and Suggestions for Authors
The case of a 34-year-old Asian male with persistent Bitot’s spots (BS) despite extensive vitamin A supplementation is described. Surgical excision of the spots followed by amniotic membrane transplantation led to symptom relief and epithelialization, with no recurrence during follow-up. Histopathologic and immunohistochemical evaluations suggested impaired cell differentiation due to vitamin A deficiency (VAD), emphasizing the need for early detection and a multidisciplinary approach in managing VAD, especially in young adults. The favorable outcome of surgical intervention prompts further investigation to optimize therapeutic strategies for VAD-related ocular manifestations.
Overall, this is an interesting case and the reviewer demands some changes for improvement and clarity of the manuscript:
A clear structure of the manuscript into Introduction, Result, Discussion, and Methods part would improve the manuscript. Importantly, a Material and Method section (immunohistochemistry) is lacking.
The corresponding email address does not agree with the indicated corresponding author’s email address? Why is another email address used for correspondence than the authors email addresses?
Line 33: Indicate for the general readership, in which panel (A or B) of Figure 1 the BS were detected.
Line 39: Did serum vitamin A levels increase after vitamin A supplementation? This would be an important
Fig. 2A: The scale bar is barely detectable and the font size of its numbering are not readable. Arrow heads pointing to the described pathologies would guide the reader and make it more clear where the pathologies are located.
Line 60: “with scattered material positive for mucin-5AC (MUC5AC)”…line 63; “and was negative for CK10, loricrin, and MUC5AC (Figure 2)”. The two statements are obviously contradictory.
Can the authors exclude that other causes than VAD contributed to the BS in this patient? For instance genetics could drive the lack of therapeutic success upon vitamin A supplementation.
Is there any reported evidence that Goblet cell differentiation in the eye’s surface depend on retinoid signaling?
Author Response
Reviewer #1: Manuscript number: diagnostics-2717892
The case of a 34-year-old Asian male with persistent Bitot’s spots (BS) despite extensive vitamin A supplementation is described. Surgical excision of the spots followed by amniotic membrane transplantation led to symptom relief and epithelialization, with no recurrence during follow-up. Histopathologic and immunohistochemical evaluations suggested impaired cell differentiation due to vitamin A deficiency (VAD), emphasizing the need for early detection and a multidisciplinary approach in managing VAD, especially in young adults. The favorable outcome of surgical intervention prompts further investigation to optimize therapeutic strategies for VAD-related ocular manifestations. Overall, this is an interesting case and the reviewer demands some changes for improvement and clarity of the manuscript: A clear structure of the manuscript into Introduction, Result, Discussion, and Methods part would improve the manuscript. Importantly, a Material and Method section (immunohistochemistry) is lacking.
Response: We greatly appreciate the Reviewer’s comment. Due to the journal's format constraints as an "Interesting Images", the ability to clearly partition the text into distinct manuscript-type sections was challenging. However, we fully recognize the importance of detailing the materials and methods, particularly for the immunohistochemistry. Please note that we have now revised the text by adding a dedicated statement elucidating the materials and immunohistochemistry protocols that were used in this study.
The corresponding email address does not agree with the indicated corresponding author’s email address? Why is another email address used for correspondence than the authors email addresses?
Response: We greatly appreciate the Reviewer’s comment, and we apologize for the oversight on our part. Please note that the corresponding email address now agrees with the indicated corresponding author’s email address.
Line 33: Indicate for the general readership, in which panel (A or B) of Figure 1 the BS were detected.
Response: We greatly appreciate the Reviewer’s comment. Please note that we have now revised the sentence on Line 33 to read as follows:
". . . Bitot's spots (BS) were observed on the nasal and temporal conjunctiva in both eyes (Figure 1 A1 and A2)."
Line 39: Did serum vitamin A levels increase after vitamin A supplementation? This would be an important
Response: We greatly appreciate the Reviewer’s comment. Yes, the patient's serum vitamin A level increased from 80 IU/dL (deficient) initially to 120 IU/dL (improved but still decreased) after vitamin A supplementation. However, there was no clinical improvement in the bilateral Bitot's spots despite the increase in the vitamin A levels.
Fig. 2A: The scale bar is barely detectable and the font size of its numbering are not readable. Arrow heads pointing to the described pathologies would guide the reader and make it more clear where the pathologies are located.
Response: We greatly appreciate the Reviewer’s comment. Please note that we have now increased the scale bar font size for improved readability. Moreover, arrow heads have now been added to point to the relevant pathologies, thus helping to guide the reader.
Line 60: “with scattered material positive for mucin-5AC (MUC5AC)”…line 63; “and was negative for CK10, loricrin, and MUC5AC (Figure 2)”. The two statements are obviously contradictory
Response: We greatly appreciate the Reviewer’s comment regarding the immunohistochemical staining results of the Bitot's spots (BS) and surrounding conjunctival tissue in the impression cytology. Please note that we have now revised the manuscript to only report the staining results of the BS themselves from the impression cytology.
Can the authors exclude that other causes than VAD contributed to the BS in this patient? For instance genetics could drive the lack of therapeutic success upon vitamin A supplementation.
Response: We greatly appreciate the Reviewer’s comment. We agree that genetics could play a role. However, given the severity and extent of keratinization observed in the conjunctival epithelium, we believe prolonged vitamin A deficiency remains the most likely primary cause of the BS in this reported case. Even with vitamin A supplementation, the turnover and replacement of the markedly affected conjunctival tissue is both difficult and slow. Therefore, we suspect the BS have persisted due to the severity of the long-standing vitamin A deficiency rather than genetics or other factors.
Is there any reported evidence that Goblet cell differentiation in the eye’s surface depend on retinoid signaling?
Response: We greatly appreciate the Reviewer’s comment. Yes, there is evidence that retinoid signaling regulates goblet cell differentiation in the eye, including in the findings in a recent report by Alam et al. (2021) showing:
・Decreased conjunctival goblet cells in vitamin A deficiency
・Attenuated goblet cell numbers with disrupted RXRα signaling
These results demonstrate that retinoid signals play an indispensable role in ocular surface goblet cell homeostasis.
Please see: Alam J, Yu Z, de Paiva CS, Pflugfelder SC. Retinoid Regulation of Ocular Surface Innate Inflammation. Int J Mol Sci. 2021;22(3):1092. Published 2021 Jan 22. doi:10.3390/ijms22031092
Reviewer 2 Report
Comments and Suggestions for Authors
The manuscript is interesting. In the manuscript, abstract and image and image discussions are presented. The patient characteristics such as race, case history and medication history can be added (if needed). In Figure - 1, it was given that the patient is Asian but Asian has a different race. The specific details of any relation with genetics (if data is available from the literature) can be given.
Author Response
Reviewer #2:
The manuscript is interesting. In the manuscript, abstract and image and image discussions are presented. The patient characteristics such as race, case history and medication history can be added (if needed). In Figure - 1, it was given that the patient is Asian but Asian has a different race. The specific details of any relation with genetics (if data is available from the literature) can be given.
Response: We greatly appreciate the Reviewer’s comment. Please note that we have now added the details about the patient's race and medical history, and that we also conducted a literature review for potential genetic factors associated with this case. However, we have yet to find any clear evidence linking genetics.